# Ni(II)-catalyzed asymmetric alkenylations of ketimines

Mao Quan[1], Xiaoxiao Wang[1], Liang Wu[1], Ilya D. Gridnev[2], Guoqiang Yang [1] & Wanbin Zhang [1]

Chiral allylic amines are not only present in many bioactive compounds, but can also be readily transformed to other chiral amines. Therefore, the asymmetric synthesis of chiral allylic amines is highly desired. Herein, we report two types of Ni(II)-catalyzed asymmetric alkenylation of cyclic ketimines for the preparation of chiral allylic amines. When ketimines bear alkyl or alkoxycarbonyl groups, the alkenylation gives five- and six-membered cyclic α-tertiary allylic amine products with excellent yields and enantioselectivities under mild reaction conditions. A variety of ketimines can be used and the method tolerates some variation in alkenylboronic acid scope. Furthermore, with alkenyl five-membered ketimine substrates, an alkenylation/rearrangement reaction occurs, providing seven-membered chiral sulfamide products bearing a conjugated diene skeleton with excellent yields and enantioselectivities. Mechanistic studies reveal that the ring expansion step is a stereospecific site-selective process, which can be catalyzed by acid (Lewis acid or Brønsted acid).

[1] Shanghai Key Laboratory for Molecular Engineering of Chiral Drugs, School of Chemistry and Chemical Engineering, Shanghai Jiao Tong University, 800 Dongchuan Road, Shanghai 200240, China. [2] Department of Chemistry, Graduate School of Science, Tohoku University, Aramaki 3-6, Aoba-ku, Sendai 9808578, Japan. Correspondence and requests for materials should be addressed to G.Y. (email: gqyang@sjtu.edu.cn) or to W.Z. (email: wanbin@sjtu.edu.cn)

n recent years, the first-row late transition-metal-catalysts have attracted a lot of attention due to their low cost and abundance, meeting the requirements for sustainable chemical synthesis[1–5]. Addition of organometallic reagents to electron-deficient double bonds is one of the most powerful reactions for the construction of various molecular skeletons. To expand the viability of transition-metal catalysts for use in addition reactions of organoboron reagents and to lower the cost of transition-metal catalysts for use in industrial processes, the first-row late transition-metal-catalyzed addition of organoboron reagents to double bonds has become an attractive target[6–24]. Nickel, as a first-row transition-metal in the same group as that of palladium, has attracted much attention[25, 26]. However, nickel-catalyzed additions of organoboron reagents to electron-deficient double bonds have not been widely reported[14–24].

Chiral amines are important skeletons found in bioactive natural products and potent drugs, therefore their asymmetric synthesis has attracted much attention over the past few decades[27–29]. Chiral allylic amines are not only present in many bioactive compounds (Fig. 1a), but can also be readily transformed to other chiral amines via the functionalization of the additional olefin group. Therefore, the asymmetric synthesis of chiral allylic amines is highly desired. Herein, we report two types of Ni(II)-catalyzed asymmetric alkenylations of cyclic ketimines for the preparation of a series of chiral α-tertiary and α-secondary allylic amines under mild reaction conditions.

## Results

**Reaction discovery.** Organoboron reagents, which can be conveniently prepared from industrial materials, have been widely used in transition-metal-catalyzed addition reactions[30–36]. Pioneered by the Hayashi group, Rh-catalyzed asymmetric additions of arylboron reagents to imines, including ketimines, have been well studied[32–46]. However, the addition of alkenylboron reagents to imines for the preparation of chiral allylic amines has not been widely reported,

possibly due to their relatively low stability. Although Rh-catalyzed asymmetric additions of alkenylborons to aldimines have been studied by two groups[47–49], a general, benign, and efficient catalytic system for similar reactions using ketimines is still highly desired (Fig. 1b)[45–47]. The Pd-catalyzed addition of alkenylboron reagents to double bonds, including imines, has not been described[50–57]. This may be due to potential side-reactions, such as homocouplings, Heck reactions, and aza-Wacker-type reactions. The Zhao group recently reported a Co(II)-catalyzed asymmetric addition of alkenylboronic acids to cyclic aldimines (Fig. 1c)[58]. However, the addition of alkenylboron reagents to ketimines, especially for the first-row late transition-metal-catalysis (including nonasymmetric process), remains highly desirable. We herein describe a general and highly efficient enantioselective addition of alkenylboronic acids to ketimines catalyzed by Ni(II) (Fig. 1d, reaction i). The preparation of seven-membered chiral sulfamides via the asymmetric addition of organoboron reagents to imines has not been previously reported, possibly due to the lack of a method for the synthesis of related substrates. Our methodology provides an alternative pathway to address this issue. We report a Ni(II)-catalyzed asymmetric cascade alkenylation/ring-expansion of unstrained five-membered cyclic ketimines via an α,α-dialkenyl substituted five-membered sulfamide intermediate (Fig. 1d, reaction ii), providing a series of seven-membered cyclic α-substituted chiral sulfamides[59, 60]. We found that in our reaction, the ring expansion step is a site-selective and stereospecific process.

**Investigation of reaction conditions.** Our study began with the reaction of cyclic ketimine (**1a**) with phenyl alkenylboronic acid (**2a**) (Table 1). In the absence of ligand, no product was obtained when the reaction was heated at reflux in trifluoroethanol (TFE) for 24 h in air, and the starting materials were recovered intact. After screening various reaction conditions including solvent and reaction temperature, we found that the desired product could be obtained with excellent yield and moderate enantioselectivity by

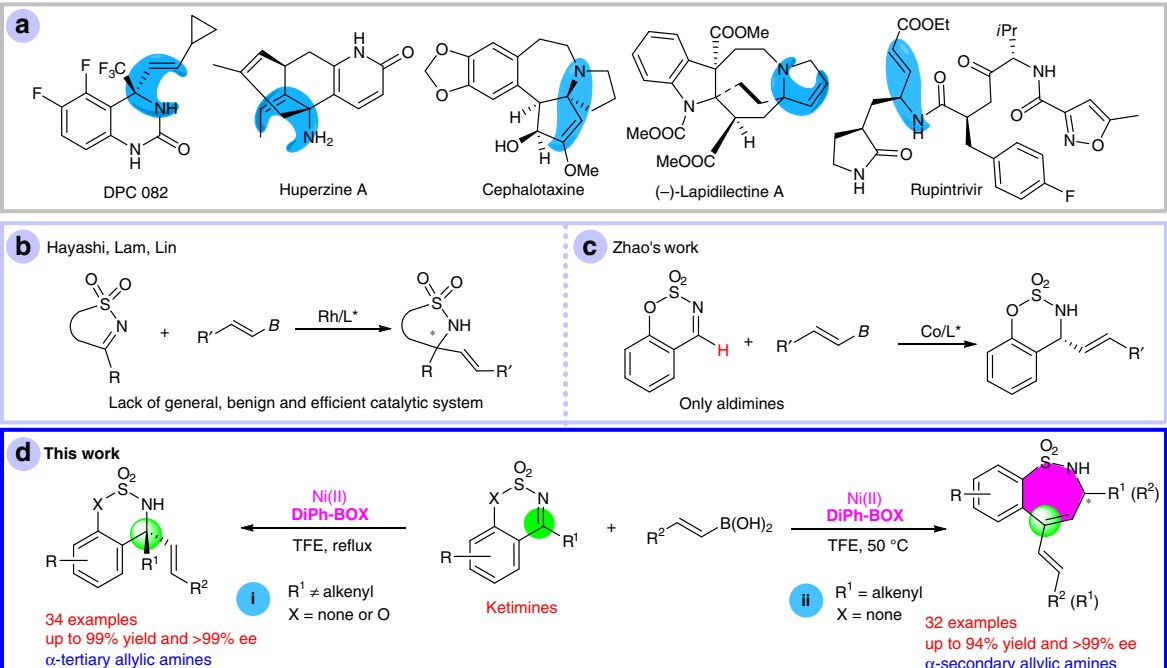

**Fig. 1** Addition of alkenylborons to imines for construction of chiral allylic amines. **a** Representative bioactive chiral allylic amines. **b** Rh-catalyzed addition of alkenylborons to ketimine. **c** Co-catalyzed addition of alkenylborons to aldimine. **d** This work: a general Ni(II)/BOX-catalyzed alkenylation and alkenylation/ring-expansion of cyclic ketimines. BOX = bisoxazoline

**Table 1 Reaction optimization**

| Entry | Ni source | Ligand | Yield %[a] | ee %[b] |
|---|---|---|---|---|
| 1 | $Ni(ClO_4)_2 \cdot 6H_2O$ | --- | NR | --- |
| 2 | $Ni(ClO_4)_2 \cdot 6H_2O$ | **L1a** | 99 | 78 |
| 3 | $Ni(ClO_4)_2 \cdot 6H_2O$ | **L1b** | 99 | 30 |
| 4 | $Ni(ClO_4)_2 \cdot 6H_2O$ | **L1c** | 99 | 90 |
| 5 | $Ni(ClO_4)_2 \cdot 6H_2O$ | **L1d** | 91 | 78 |
| 6 | $Ni(ClO_4)_2 \cdot 6H_2O$ | **L1e** | 96 | 70 |
| 7 | $Ni(ClO_4)_2 \cdot 6H_2O$ | **L1f** | 96 | 14 |
| 8 | $Ni(ClO_4)_2 \cdot 6H_2O$ | **L2** | 99 | 92 |
| 9 | $Ni(ClO_4)_2 \cdot 6H_2O$ | **L3** | 87 | 90 |
| 10 | $Ni(ClO_4)_2 \cdot 6H_2O$ | **L4** | trace | --- |
| 11 | $Ni(ClO_4)_2 \cdot 6H_2O$ | **L5** | trace | --- |
| 12 | $Ni(ClO_4)_2 \cdot 6H_2O$ | **L6** | trace | --- |
| 13 | $NiCl_2 \cdot 6H_2O$ | **L2** | 97 | 86 |
| 14 | $NiSO_4 \cdot 6H_2O$ | **L2** | 86 | 80 |
| 15 | $Ni(OAc)_2 \cdot 4H_2O$ | **L2** | 91 | 81 |
| 16 | $NiBr_2$ | **L2** | 80 | 86 |
| 17 | $NiCl_2 \cdot DME$ | **L2** | 94 | 89 |
| 18 | $Ni(acac)_2$ | **L2** | 83 | 80 |
| 19 | $Ni(OTf)_2$ | **L2** | 99 | 94 |
| 20 | --- | **L2** | NR | --- |

Reactions were carried out on a 0.20 mmol scale (**1a**) using *trans*-PhCH=CHB(OH)$_2$ (**2a**) (0.30 mmol), 5 mol% nickel salt, 7.5 mol% ligand in unpurified TFE (2.0 mL) in a test tube for 24 h which was opened to air
[a]Isolated yields
[b]Determined by HPLC using a chiral Enanticol column. TFE = trifluoroethanol, NR = no reaction, ee = enantiomeric excess

carrying out the reaction with 5.0 mol% of Ni(ClO$_4$)$_2$·6H$_2$O, 7.5 mol% bisoxazoline ligand **L1a** in TFE at reflux for 24 h, under air (Table 1, entry 2. See Supplementary Table 1). The screening of substituents on the oxazoline rings of the ligand showed that the phenyl group gave the best results (see Supplementary Table 1). The effect of substituents on the bridge of the BOX ligand was also studied, and the best ee was obtained with ligand **L1c** bearing *n*-Pr groups (Table 1, entries 2–7). An additional phenyl group on

**Table 2 Scope of ketimine substrates**

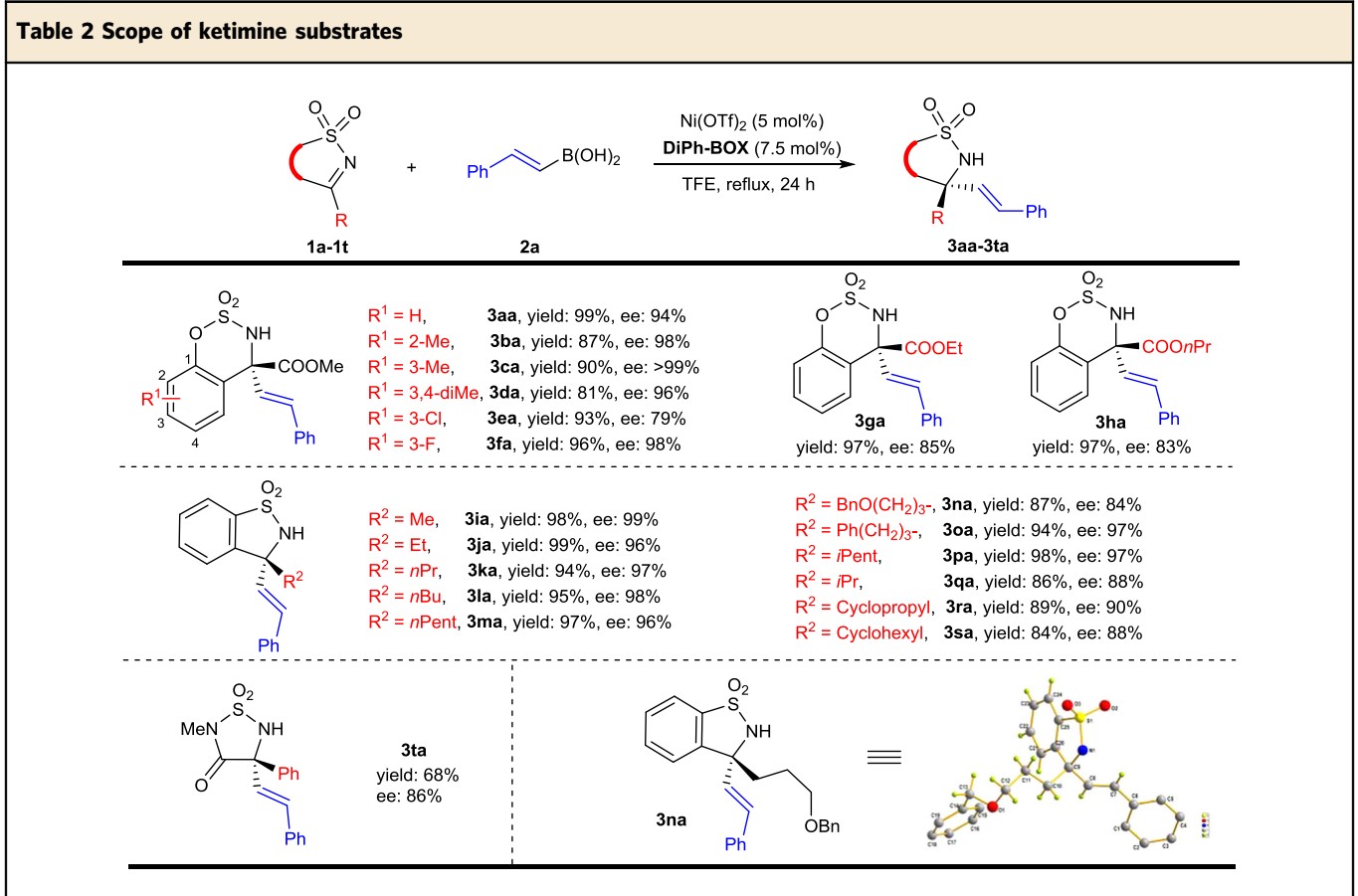

Reactions were carried out on a 0.20 mmol scale using 5 mol% Ni(OTf)₂, 7.5 mol% ligand and *trans*-PhCH=CHB(OH)₂ (**2a**) (0.30 mmol) in TFE (2.0 mL) at reflux in a test tube for 24 h, which was opened to air. The yields are isolated yields and ee's were determined by chiral HPLC

both of the oxazoline rings (**L2 = DiPh-BOX**) led to an increase in ee from 90% to 92% (Table 1, entry 8 vs entry 4). Reactions with other types of ligands showed moderate or poor results (**L3–L6**, Table 1, entries 9–12). Finally, Ni(OTf)₂ was found to be the best nickel source for the alkenylation (Table 1, entry 19). All other nickel salts resulted in a slight decrease in yield or ee (Table 1, entries 13–18). None of the desired product was obtained in the absence of Ni salt (Table 1, entry 20).

**Scope of the asymmetric alkenylation of ketimines**. With the optimized conditions in hand, the ketimine substrate scope was examined (Table 2). Substrates possessing methyl groups at different positions gave slightly higher ee's (**3ba–3da** vs **3aa**). Fluoride and chloride substituents have opposite effects on enantioselectivity (**3ea** and **3fa**). Lower ee's were obtained for substrates bearing bulkier ester groups (**3ga** and **3ha**). To our delight, saccharin-derived ketimines are also compatible substrates for this reaction (**3ia–3sa**). Branched alkyl-substituted substrates provided the corresponding products with slightly lower ee's and yields when compared to their linear alkyl-substituted counterparts (**3ia–3sa**). Our catalytic system could also be applied to substrate **1t**[46] giving a chiral α-tertiary amino acid derivative. The absolute configuration of the alkenylation product **3na** was determined to be (*R*) by X-ray crystallographic analysis (Table 2).

The alkenylboronic acid scope was also investigated, the results of which are summarized in Table 3. Firstly, substrate **1i** was used as a ketimine for reaction with different alkenyl nucleophiles. The type of substituents on the benzene ring of the styrene-boronic

acids have little influence on the results, and excellent yields and enantioselectivities could be obtained (**3ia–3if**). It is worth noting that the reaction proceeded smoothly when aliphatic group-substituted vinyl boronic acids were used as nucleophiles (**3ig–3il**). Disubstituted and trisubstituted vinyl boronic acids gave their corresponding products with good to excellent yields and enantioselectivities (**3ig–3il**). Substrate **1a** was also tested as a substrate for the addition of several different styrene-derived boronic acids, and good results were obtained (**3ab–3ad**).

**Ni(II)-catalyzed asymmetric alkenylation/ring-expansions**. We were also interested in studying the regioselective and enantioselective issues of the addition of alkenylboronic acids to α,β-unsaturated cyclic ketimines. To our surprise, a seven-membered α-substituted chiral sulfamide (**5a**) bearing a conjugated diene system was obtained (Table 4). We next focused on developing this interesting asymmetric (*n* + 2)-ring expansion reaction.

Three- and four-membered vinyl cyclic compounds have been well studied for the construction of five- and six-membered systems via (*n* + 2)-expansion due to their strong tendency to release their high ring strain[61, 62]. However, the (*n* + 2)-expansion of larger vinyl cyclic compounds remains challenging due to the fact that they possess less ring strain than their smaller ring counterparts, an essential driving force for ring expansion[63]. Furthermore, the asymmetric synthesis of chiral compounds via this ring (*n* + 2)-expansion has not been widely reported, possibly due to the lack of interactions between the alkenyl group and chiral catalyst that would allow for the remote control of the attacking enantioface at the β-carbon of the C=C bond[64–66].

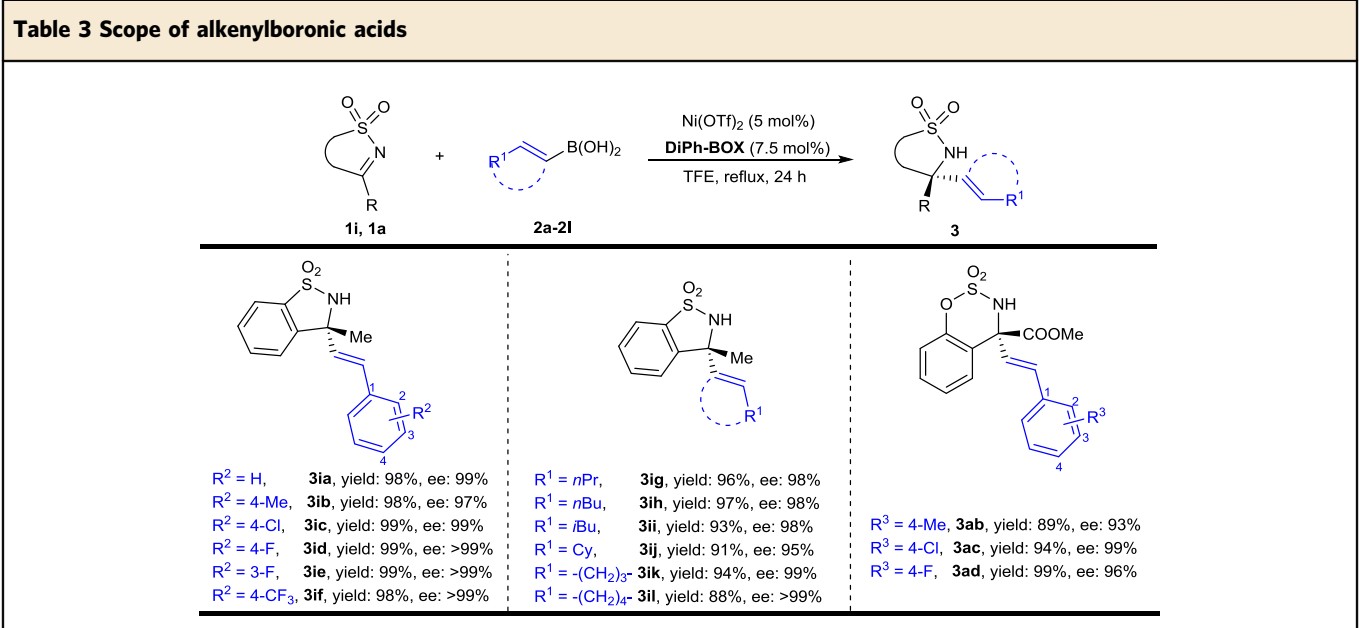

**Table 3 Scope of alkenylboronic acids**

Njardarson reported a chiral acid-catalyzed asymmetric ring ($n$ + 2)-expansion of dialkenyl oxetane with only one substrate, requiring a low temperature and long reaction time to obtain the desired product with good ee and moderate yield[67]. Conversely, our catalytic system enables the ring-expansion of unstrained five-membered aza-rings (Table 4), providing a series of seven-membered α-substituted chiral sulfamides[59, 60].

**Scope of the asymmetric alkenylation/ring-expansions**. After screening a series of parameters, the optimal reaction conditions were found to be when using Ni(OTf)$_2$ as nickel source, **DiPh-BOX** (**L2**) as a chiral ligand, and performing the reaction in TFE solvent at 50 °C over 24 h (see Supplementary Table 2). It should be noted that other types of ligand were ineffective for this reaction. With the optimized conditions in hand, the alkenylboronic acid scope was examined (Table 4). Two series of catalytic reactions were carried out: the reaction of alkyl vinyl-substituted ketimines with styrenyl boronic acids (Table 4, Parts i–iii); and the reaction of styrenyl ketimines with alkyl-substituted vinyl boronic acids (Table 4, Part iv). Interestingly, the results showed that the site-selectivity of this cascade alkenylation/ring-expansion could be controlled by the driving force for the formation of a larger π-conjugated system, thus only isomer A was observed while none of the B type isomer was detected in the crude $^1$H NMR spectra (Table 4, Part v).

Firstly, different styrenyl boronic acids were tested with substrate **4a** (Table 4, Part i). The results showed that styrenyl boronic acids bearing either electron-donating or electron-withdrawing substituents on the benzene ring gave the corresponding products with good to excellent yields and ee's (**5a**–**5j**). The enantioselectivities for substrates bearing electron-donating substituents are comparatively lower than those for substrates bearing electron-withdrawing substituents, albeit good ee's could still be obtained for all examples (**5b**–**5d** vs **5e**–**5j**). The absolute configuration of **5f** was confirmed to be (*S*) by X-ray crystallographic analysis (Table 4, Part vi).

Substrates with different alkyl vinyl substituents were also investigated and the results are summarized in Part ii of Table 4. When R$^1$ is alkyl and R$^2$ is aryl, the corresponding products **5k**–**5p** were obtained with excellent yields and ee's. Substituents on

the benzene ring of the substrates have little influence on the reaction outcome (Part iii, **5q**–**5s**).

The R$^1$ group does not necessarily end up on the carbon atom α to the nitrogen of the product. When R$^1$ was aryl and R$^2$ was alkyl, the products **6**, possessing the opposite configuration of **5** and with the R$^2$ group at the α-position of the nitrogen, were obtained (Table 4, Part iv). Different substrates gave their corresponding products with good to excellent ee's. The desired products were obtained with lower yields and required longer reaction times (**6a**–**6j**) compared to that of products **5**, most likely due to the poor solubility of the substrates. Product **6k** bearing a thienyl substituent was obtained in moderate yield and ee. The absolute configurations were confirmed to be opposite by comparison of the HPLC spectra and optical rotation data of **6c** and (*R*)-**5p**.

Surprisingly, when linear alkyl-substituted vinylboronic acids were used for reaction with **4a**, the expansion of the ring occurred on the linear alkyl group side (Fig. 2), perhaps as a result of the steric hindrance of the bulky *t*Bu group. Good yields and ee's could also be obtained in both cases.

**Mechanistic considerations**. The results above indicate that nickel is capable of catalyzing the asymmetric alkenylation and alkenylation/ring-expansion of cyclic ketimines. The importance of this process prompted us to investigate the reaction mechanism. We propose that (*S*)-**7a** is the intermediate of the cascade reaction, which is the product resulting from alkenylation of the ketimine. Firstly, the stereochemistry and catalyst for the ring expansion step were studied using control experiments[48, 49]. Compound (*S*)-**7a** was stirred in TFE at 50 °C for 24 h. In the absence of catalyst and alkenylboronic acid, no ring expansion was observed (Fig. 3a). When the reaction was conducted with catalyst but no alkenylboronic acid, the product **5a** was obtained with 49% yield and with no loss in ee (Fig. 3b). Interestingly, ring expansion product **5a** could also be obtained in the presence of only the alkenylboronic acid or using (MeO)$_2$P(O)OH as a catalyst, without any loss in ee (Fig. 3c, d). Furthermore, when ring expansion of (*R*)-**7a** was catalyzed by Ni catalyst, the enantiomeric product **8** was obtained (Fig. 3e). The above results indicate that the ring expansion step is an acid-promoted process, either

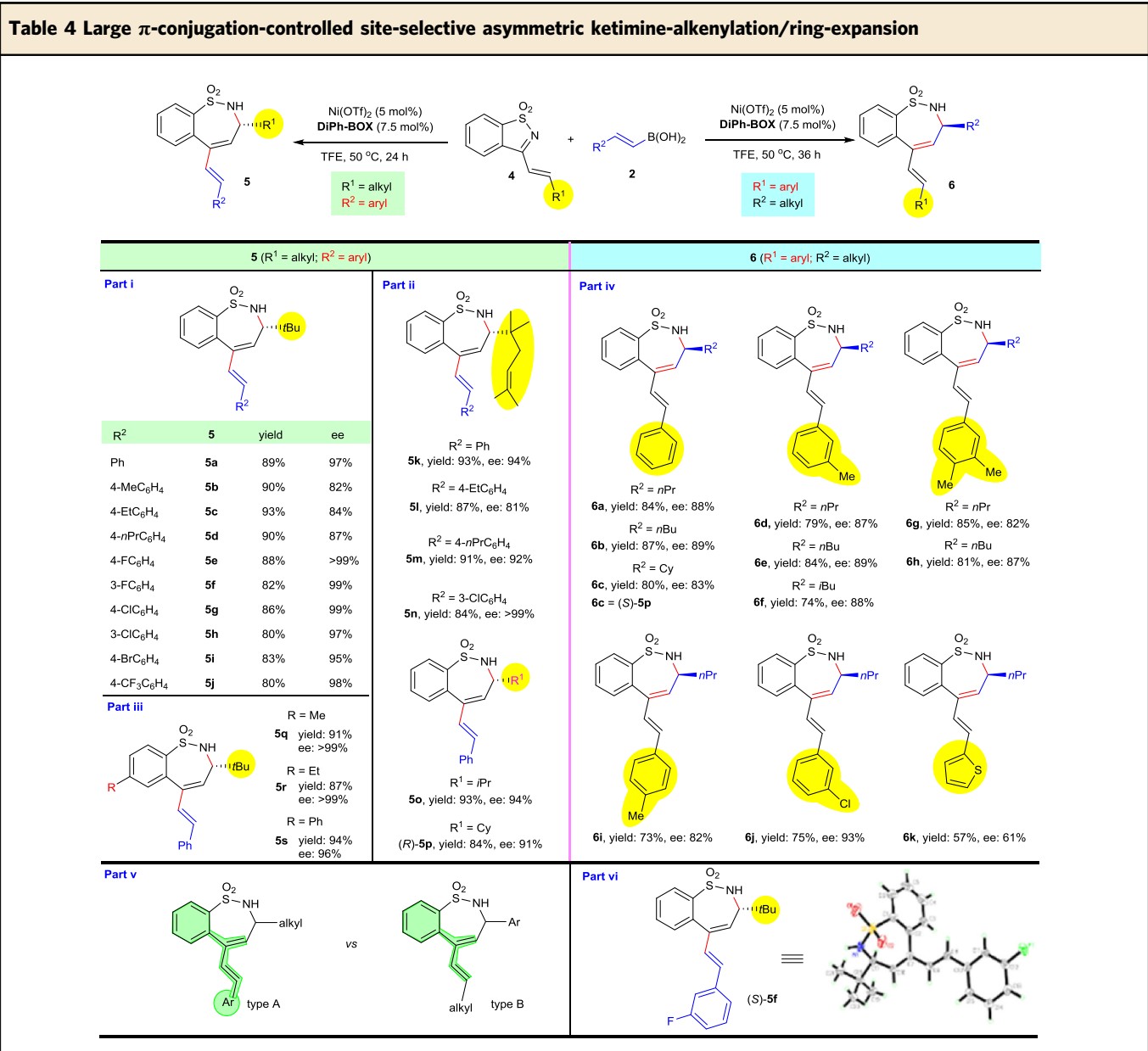

**Table 4 Large π-conjugation-controlled site-selective asymmetric ketimine-alkenylation/ring-expansion**

**Fig. 2** Sterically-controlled site-selective asymmetric alkenylation/ring-expansion of ketimine **4a**. The results using linear alkyl-substituted vinylboronic acids as nucleophiles reaction with **4a** indicate that the asymmetric alkenylation/ring-expansion can also be controlled by steric hindrance

by a Lewis acid or Brønsted acid[59, 60, 64–66], and no matching/mismatching effect was observed between the catalyst and intermediate **7a**. The ring expansion step for this reaction is a stereospecific process (see Supplementary Note 2 for preliminary DFT calculation results), suggesting that the chiral induction by the ligand occurs at the ketimine-alkenylation step, which is a regio-specific 1,2-addition process. It should be noted that Xu has found that the Rh-catalyzed asymmetric addition of arylboron to

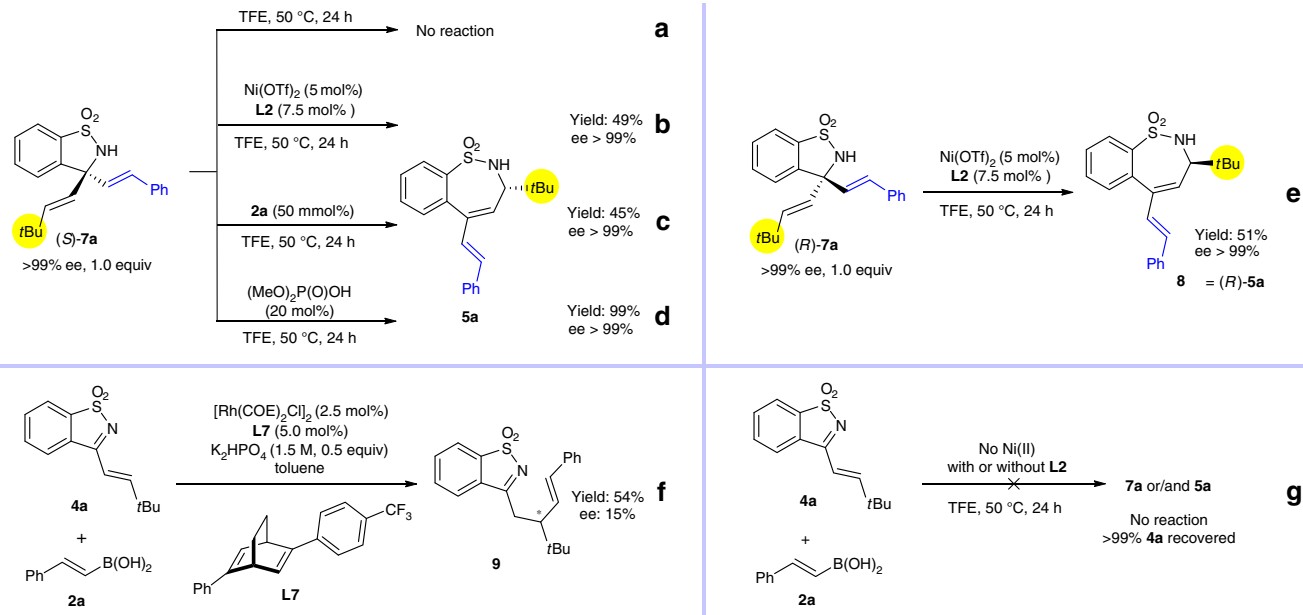

**Fig. 3** Control experiments. These experiments indicate that the ring expansion step is an acid-promoted stereospecific process

**Fig. 4** Model for explaining the origin of asymmetric induction. The alkenyl attacks the C=N bond from the *Re*-face to generate the products (*R*)-**3na** or (*S*)-**7a**

α,β-unsaturated cyclic ketimines gave regioselective 1,4-addition products[68]. The 1,4-addition product **9** was also obtained for the reaction of **4a** and **2a**, using rhodium as a catalyst (Fig. 3f). These reactions indicate the unique property of Ni-catalysts for addition reactions with alkenylborons compared to Rh-catalysts[58, 69]. Two control experiments were also conducted. Reactions of **4a** with **2a** in the absence of Ni(II) and with or without ligand **L2**, did not give rise to product **7a** or **5a**, with **4a** being completely recovered (Fig. 3g). This also indicates that the alkenylation step requires the assistance of Ni(II) and the chiral ligand.

The stereochemical outcome can be explained using the model shown in Fig. 4, similar to that of Rh- and Pd-catalysis[30–36, 50–57]. The substrate coordinates with nickel and the sulfonyl group lies on the opposite side of diphenyl substituents on the oxazoline ring. This leads to (*R*)-**3na** as the major product and also leads to (*S*)-**7a** as the intermediate for producing **5a**.

For α,α-dialkenyl heterocyclic compounds bearing two different alkenyl groups, control of the site-selectivity of the ring expansion and maintaining optical purity are challenging (Fig. 5a). Since **7** was confirmed to be the intermediate of the reaction for the preparation of seven-membered sulfamides from five-membered ketimines, our research shows that the aforementioned challenges can be overcome by choosing suitable substrates. A mechanistic explanation of this site-selective stereospecific process has been proposed (Fig. 5b). Ketimine substrate **4** undergoes Ni(II)/BOX-catalyzed addition of alkenylboronic acid to form dialkenyl sulfamide intermediate

**7** following a mechanism similar to that shown in Fig. 4. Intermediate **7** is then activated immediately by the acid (Lewis acid or Brønsted acid) and the C–N bond is cleaved to generate a carbocation intermediate. When alkyl vinyl ketimine and styrenyl boronic acid are used (R[1] = alkyl, R[2] = aryl; Fig. 5b), carbocation model A is generated. When styrenyl ketimine and alkyl vinyl boronic acid are used (R[1] = aryl, R[2] = alkyl; Fig. 5b), carbocation model B is generated. In models A and B, the sulfamide anion attacks the alkyl-substituted carbon to yield the large π-conjugated ring system **5** or **6**, respectively, with opposite configurations. When R[1] is a *t*Bu group and R[2] is a *n*Pr group, the sulfamide anion attacks the less sterically hindered carbon atom of model C, yielding **6l**. The rotation of the single bond between the sulfonyl benzene and carbocation can lead to lower ee's for the desired products (racemization process), however, it seems that this racemization process does not occur easily under the optimized reaction conditions due to that steric hindrance exists around the axis of this single bond (like that observed for axially-chiral biphenyl compounds) (see Supplementary Note 2 for preliminary DFT calculation results). In consideration of the experimental results shown in Table 4, it can be concluded that this reaction favors larger π-conjugation over lower steric hindrance.

**Synthetic utility of alkenylation products.** A gram-scale reaction was conducted and the product **3ca** was obtained in good yield with excellent enantioselectivity (Fig. 6a). The ester group of **3ca**

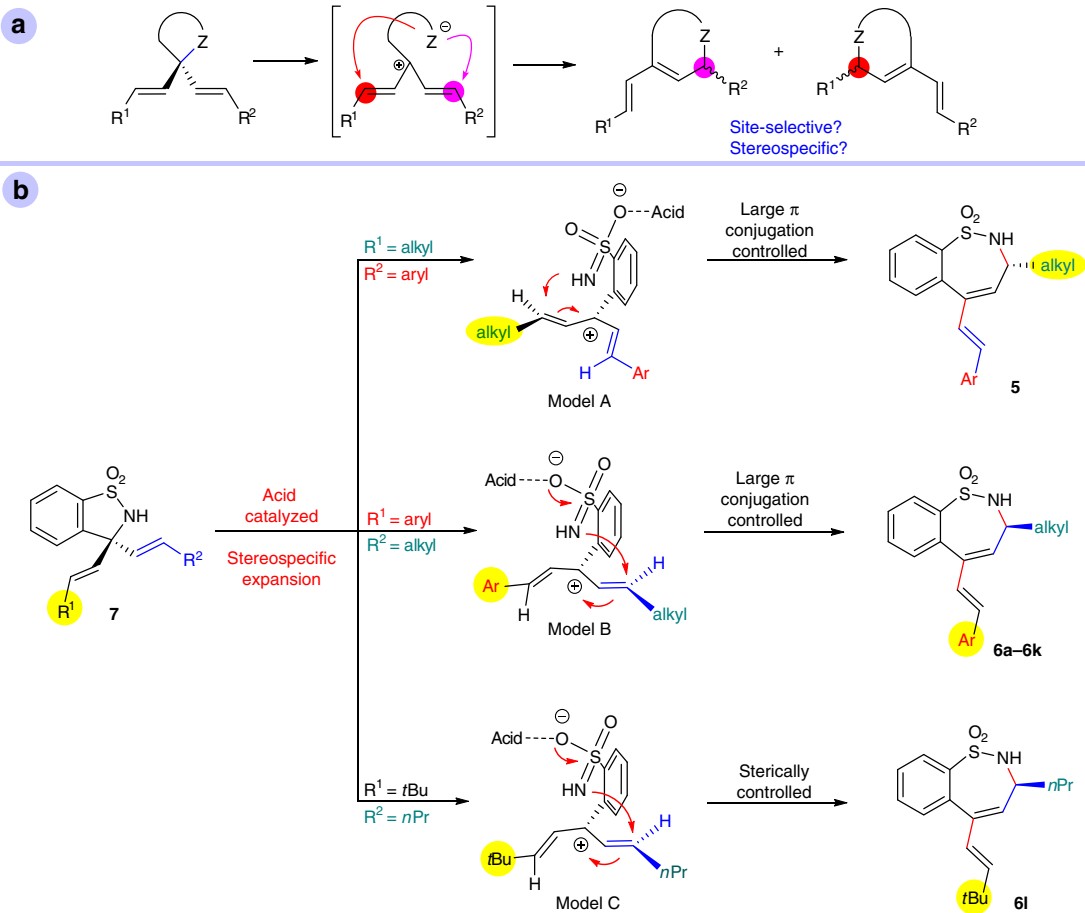

**Fig. 5** Challenges for ring-expansion and proposed mechanism. **a** Challenges for ring-expansion of α,α-dialkenyl heterocyclic compounds bearing two different alkenyl groups. **b** Proposed mechanism for ring-expansion step

was reduced with LiBH$_4$ conveniently to **10**. A bromoamination of the olefin group of **10** then gave cyclization product **11**, a chiral aziridine bearing three carbon stereocenters, including a tetra-substituted carbon atom. Product **3aa** was reduced to a tetra-substituted amino alcohol by LiAlH$_4$ with excellent yield and enantioselectivity (see Supplementary Note 3). The alkenylated amino acid product **3ta** was also converted to interesting skeletons (Fig. 6b). After allylation of the nitrogen of the sulfamide, a metathesis reaction catalyzed by Grubbs-II catalyst was carried out, which furnished a chiral 2,2-disubstituted pyrrolidine **13** without any loss in ee. This skeleton, which contains at least one aryl substituent, represents an important class of pharmaceutically relevant compounds[70].

Several transformations of seven-membered sulfamide product **5a** were also performed (Fig. 6c). Both C=C bonds of **5a** could be hydrogenated using Pd/C as a catalyst. Product **14** was obtained with no loss in ee but with moderate dr (diastereoisomer ratio). Interestingly, when [Rh(iPr-Ferphos)(cod)]BF$_4$ was used as a catalyst, the disubstituted C=C bond could be hydrogenated selectively to generate **15** with no reduction in ee. A Diels–Alder reaction was also conducted. Firstly, **5a** was quantitatively methylated to **16** with MeI. Subsequently, the corresponding benzyne was generated in situ from 2-(trimethylsilyl)phenyl trifluoro-methanesulfonate, which was reacted with **16** to give the tetracyclic compound **17** with 5:1 dr and 97% ee.

## Discussion

In summary, we have developed a general enantioselective addition of alkenylboronic acids to ketimines enabled by a Ni(II)/

BOX-catalytic system. A relatively wide substrate scope of cyclic ketimines and alkenylboronic acids are compatible with our reaction conditions. The desired chiral α-tertiary allylic amines can be obtained in good yields (up to 99%) and enantioselectivities (up to >99% ee). A Ni(II)-catalyzed asymmetric cascade alkenylation/ring-expansion reaction of alkenyl cyclic ketimines was also developed. A series of seven-membered chiral sulfamides were synthesized with good yields (up to 94%) and enantioselectivities (up to >99% ee) under mild reaction conditions. This reaction is also an efficient method for the preparation of trisubstituted conjugated dienes. Mechanistic studies showed that the alkenylation is the enantioselectivity-determining step, while the ring expansion step is a stereospecific process. The site-selectivity of the ring rearrangement expansion can be controlled by the formation of a large π-conjugated system or by steric interactions. Transformations of products for both types of reactions were conducted to show the potential applications of our methodologies.

## Methods

**Procedure for asymmetric alkenylations of ketimines**. A test tube (100 mL) was charged with Ni(OTf)$_2$ (3.6 mg, 0.010 mmol, 0.050 equiv), **L2** (8.1 mg, 0.015 mmol, 0.075 equiv), and unpurified TFE (1.0 mL). The solution was stirred at reflux for 5 min, then substrate (0.20 mmol, 1.0 equiv) and alkenylboronic acid (0.30 mmol, 1.5 equiv) were added into the tube. The wall of the tube was rinsed with an additional portion of TFE (1.0 mL). After stirring at reflux for 24 h in air, the reaction mixture was cooled to room temperature and the solvent was removed by rotary evaporation. The residue was purified by preparative TLC on silica gel (petroleum ether/EtOAc = 5/1) to give the product.

**Procedure for asymmetric alkenylation/ring-expansions**. A test tube (100 mL) was charged with Ni(OTf)$_2$ (3.6 mg, 0.010 mmol, 0.050 equiv), **L2** (8.1 mg, 0.015

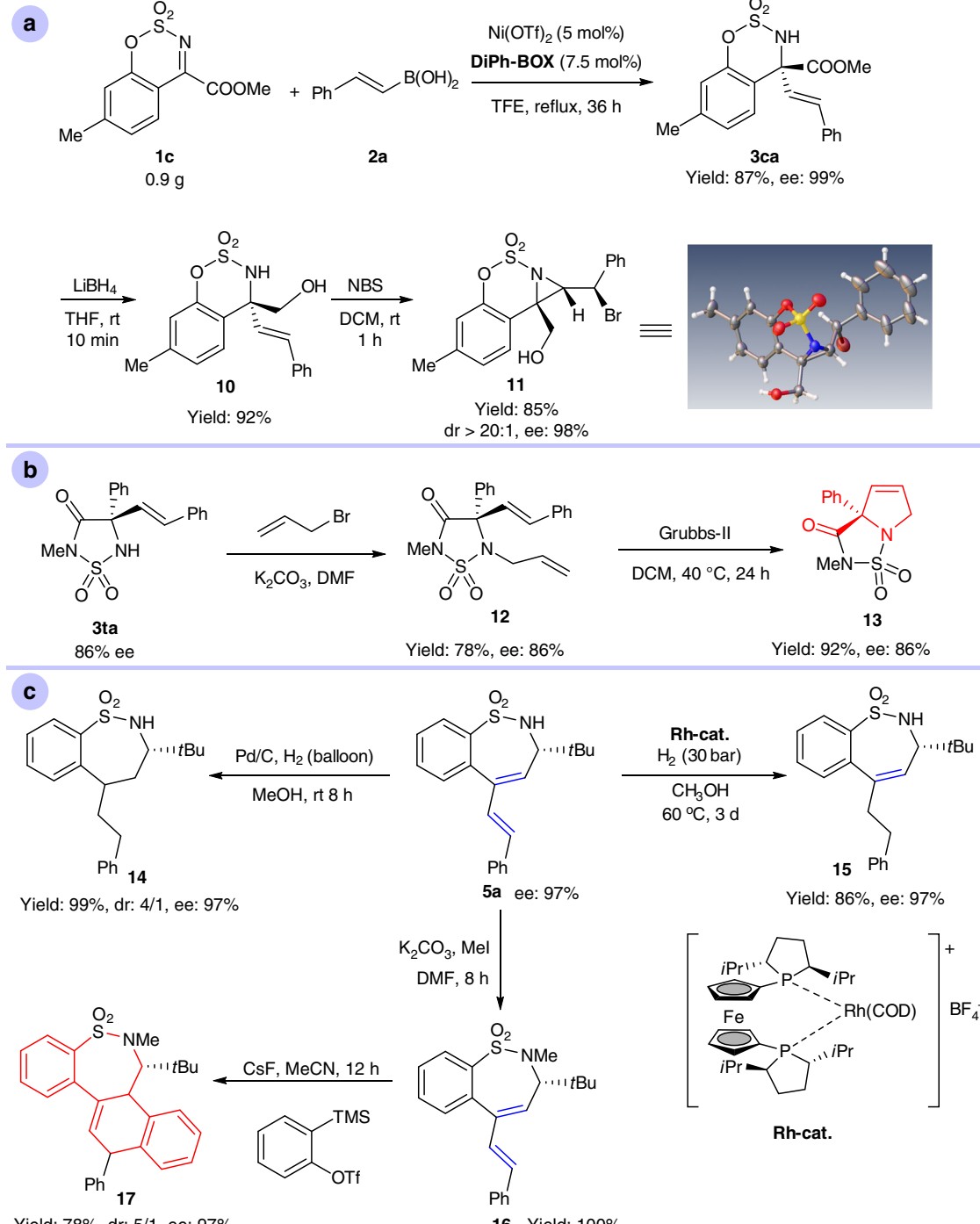

**Fig. 6** Synthetic utility of alkenylation products. **a** Gram-scale reaction and transformations of **3ca**. **b** Transformations of **3ta**. **c** Further transformations of **5a**

mmol, 0.075 equiv), and unpurified TFE (1.0 mL). The solution was stirred at 50 °C for 20 min, then substrate (0.20 mmol, 1.0 equiv) and alkenylboronic acid (0.30 mmol, 1.5 equiv) were added into the tube. The wall of the tube was rinsed with an additional portion of TFE (1.0 mL). After stirring at 50 °C for 24 h or 36 h in air, the reaction mixture was cooled to room temperature and the solvent was removed by rotary evaporation. The residue was purified by preparative TLC on silica gel (petroleum ether/EtOAc = 5/1) to give the product.

CCDC 1486078, **5f**: CCDC 1556426, **11**: CCDC 1816920). These data could be obtained free of charge from The Cambridge Crystallographic Data Centre via www.ccdc.cam.ac.uk/data_request/cif.

**Data availability**. The authors declare that the data supporting the findings of this study are available within the article and its Supplementary Information file. For the experimental procedures, and NMR and HPLC analysis of the compounds in this article, see Supplementary Methods and charts in the Supplementary Information file. The X-ray crystallographic coordinates for structures reported in this article have been deposited at the Cambridge Crystallographic Data Centre (**3na**:

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

## Acknowledgments

We thank the National Natural Science Foundation of China (Nos. 21302124, 21620102003, and 21772119), Science and Technology Commission of Shanghai Municipality (No. 15JC1402200), and Shanghai Municipal Education Commission (No. 201701070002E00030) for the financial support.

## Author contributions

M.Q. conducted most of the synthetic experiments. X.W. and L.W. conducted part of the synthetic experiments. I.D.G. conducted the DFT computational study. W.Z., G.Y., and M.Q. wrote the manuscript. G.Y. and W.Z. directed the project. All authors contributed to discussions.

## Additional information

**Competing interests:** The authors declare no competing interests.

