## [Peer Review File · Nature Communications]

Reviewer #1 (Remarks to the Author):

This manuscript describes the development of enantioselective, nickel-catalyzed alkenylations of cyclic ketimines to deliver vinyl-substituted alpha-tertiary amines, and the subsequent ring-expansion of some of these products to form enantioenriched 7-membered rings. This alkenylation builds upon previous reports mainly using rhodium-based catalysts for arylation of similar substrates, but makes advances both in the use of a first-row, nonprecious base metal (nickel) and in performing alkenylation instead of arylation. It also tackles the challenging enantioselective formation of tetrasubstituted stereocenters, for which limited methods are available. The conditions are mild (even performed open to air) and use an easily prepared ligand (1 step from commercial materials). Both 5- and 6-membered ketimines work in this chemistry, and reasonable functional group tolerance is observed, providing good scope for this reaction. In addition, the authors demonstrate that the reaction can be run on gram-scale and derivatize a number of products to show their versatility.

The most exciting aspect of this work is the serendipitously discovered ring-expansion of the 5-membered allylic amines to 7-membered rings. Although ring expansions are known, they typically harness release of ring strain as a driving force, and this is not the case here. These medium-sized rings are valuable, and their formation under these conditions is both surprising and mechanistically intriguing. The authors show that the ring expansion is stereospecific, and occurs in the presence of either the Ni catalyst or boronic acid (Fig 5). Based on these experiments, they reasonably hypothesize that the expansion is Lewis acid-catalyzed (or mediated). This leads them to the proposal that the expansion proceeds via a pentadienyl carbocation, and that the benzyl ring does not rotate in this carbocation intermediate (Fig 8). This proposal is highly speculative (and will likely be controversial); it is unclear why the benzyl ring would not rotate into conjugation with this highly stabilized (and thus long-lived) carbocation. The language should be toned down in how firmly the authors propose this provocative mechanism. With that change, and the other minor revisions noted below, I recommend publication.

Minor revisions:

- In the mechanism discussion, it would be helpful to include a brief mechanism proposal for the vinylation step.
- The authors report a number of ee's with decimal places (ex: 99.8% ee). Although the authors clearly measured this ee, I do not believe it is appropriate to include this level of accuracy, unless these reactions have been run in duplicate or triplicate, and this ee is reproducible to this level of accuracy. >99% ee is more reasonable.
- What is the stereochemistry of product 9?
- For the experiment in Fig 5 with Ni(OTf)₂/L2, is there a match/mismatch between the catalyst and diene 7? Does the yield or ee change if the opposite enantiomer of 7 is used?
- There are a number of typos throughout the paper and Supporting Information.
- In the Supporting Information:
 - o How were racemic samples prepared? A brief note will suffice.
 - o Was a Pybox ligand tried for the vinylation (page 6 of the SI)?
 - o In the optimization studies, how was reflux maintained without loss of solvent in an open test tube? An experimental for these optimization studies would be helpful.

- o In a number of the HPLC traces of enantioenriched compounds, there is more than one minor peak near the retention time from the racemic trace (page 130, 166, 167, 188). How did the researchers know which peak to integrate? A co-injection of racemic and enantioenriched compound may be necessary to correctly assign the minor peak in these cases.
- o On page 183, the labels for the traces are confusing. It is recommended that the reaction 5-membered ring + catalyst → 7-membered ring is shown (not just reagents). Also, the HPLC trace of the racemic 7-membered ring should be added to this page for easy reference.

Reviewer #2 (Remarks to the Author):

Quan et al. reported new nickel catalysis that accomplished asymmetric alkenylation of ketimines. The authors optimized the catalyst with reactive electron-deficient 6-membered ketimine 1a. They found L2 as the best ligand in terms of yield and enantioselectivity. Notably, Zhao commented the limitation on cobalt catalysis (JACS 2016, 138, 6571) that "The corresponding ketoimine substrates bearing a methyl or ester substituent (11f and 11g, Table 2) were also examined under the optimal conditions. Unfortunately, no reactivity was observed, which clearly points to the limitation of the catalytic activity of this cobalt-catalyzed system." The present nickel-catalyzed reaction thus overcomes the limitation of asymmetric alkenylation of 6-membered ketimines. However, the reaction of alkyl- or alkenyl substituents (at the carbon atom of C=N bond) in 6-membered ketimines were not presented and the scope is mainly focused on saccharin derivatives and ester-substituted 6-membered ketimines. Although this work presented substantial improvement on nickel-catalyzed reaction, the impact is somehow limited in asymmetric catalysis field, which dampens my enthusiasm for this being published in Nature Commun.

It is interesting that the authors found surprising ring-expansion during the study on substrate scope (Table 4). The reaction of ketimines having alkenyl group gave 7-membered products 5 or 6. The ring-expansion took place with the interaction of Lewis acid (nickel or boronic acid). They studied the mechanism of this ring expansion, which was supported with some experiments. The utility of the products was shown in Figs 9-11, exhibiting the potential importance of these products. However, I think these results cannot attract broader interests from chemists who are not working on asymmetric catalysis as well as other non-organic chemists who are the scope of this journal.

All together, I think this paper is more suited to a chemistry journal such JACS as an appropriate media.

Below are some specific comments that the authors should address as these would be beneficial for improvement:

- 1) Is this reaction applicable with other nucleophiles such as arylboronic acid?
- 2) Why this nickel-catalyzed reaction shows high selectivity of 1,2-addition rather than 1,4-addition (of rhodium catalyst)? Is this selectivity controlled (or chanted) by the ligand? Or just nickel has tendency to undergo 1,2-addition?

3) There are many typos. When submitting the manuscript, the authors should be more careful.

line 27, Nickle should be Nickel

line 29, nickle

line 35 additioanl should be additional

lin 37 prepration should be preparation

Fig.2. in c) this work, R"1" should be superscript.

line 226 alkneyl should be alkenyl

line 227 mantaining should be maintaining

line 252 reduced might be reduced

line 525 Natrual Scicence should be Natural Science

Response to Referees

Thank you very much for your work on our manuscript. The modifications for our manuscript are as follows:

Referee 1:

- 1) The authors show that the ring expansion is stereospecific, and occurs in the presence of either the Ni catalyst or boronic acid (Fig 5). Based on these experiments, they reasonably hypothesize that the expansion is Lewis acid-catalyzed (or mediated). This leads them to the proposal that the expansion proceeds via a pentadienyl carbocation, and that the benzyl ring does not rotate in this carbocation intermediate (Fig 8). This proposal is highly speculative (and will likely be controversial); it is unclear why the benzyl ring would not rotate into conjugation with this highly stabilized (and thus long-lived) carbocation. The language should be toned down in how firmly the authors propose this provocative mechanism.

Response: We have conducted some preliminary DFT calculations to study the rotation process (see SI). The results showed that once the charge-separated intermediate formed, the N anion readily attacks the carbon near the *t*Bu group with very little in the way of a transition state. This means that the second step (C-N formation) is very fast step. However, if the benzyl ring of the intermediate rotates into conjugation with the pentadienyl carbocation, there is steric hindrance around the C-Ar(SO₂N) axis (~12 kcal/mol higher), like that observed for axially-chiral biphenyl compounds, therefore, the carbocation proposed by this reviewer is not long-lived. This rotation would result in racemization, but it is an unfavored process due to steric-hindrance. Thus, the expansion step is a stereospecific process.

- 2) In the mechanism discussion, it would be helpful to include a brief mechanism proposal for the vinylation step.

Response: We have provided a brief mechanism proposal in the text (Fig. 6) and the description following Fig. 6.

- 3) The authors report a number of ees with decimal places (ex: 99.8% ee). Although the authors clearly measured this ee, I do not believe it is appropriate to include this level of accuracy, unless these reactions have been run in duplicate or triplicate, and this ee is reproducible to this level of accuracy. >99% ee is more reasonable.

Response: Thanks for providing this suggestion. We have changed the ees to >99%, instead of the former description.

- 4) What is the stereochemistry of product 9?

Response: Thanks for providing this very important suggestion. We should have been more careful. We have obtained the single crystal structure of 9 (the number in the revised manuscript has been changed to 11). The structure showed that it is not a tetrahydrofuran derivative, but an interesting chiral aziridine derivative.

- 5) For the experiment in Fig 5 with Ni(OTf)₂/L2, is there a match/mismatch between the catalyst and diene 7? Does the yield or ee change if the opposite enantiomer of 7 is used??

Response: We have conducted this experiment and no matching/mismatching effect was observed between the catalyst and intermediate 7. These results have been described in the revised manuscript.

- 6) There are a number of typos throughout the paper and Supporting Information.

Response: We have revised the typos both in the paper and SI.

7) How were racemic samples prepared? A brief note will suffice.

Response: We have added related notes to the SI.

8) Was a Pybox ligand tried for the vinylation (page 6 of the SI)?

Response: We have tried this experiment but there was no reaction under our conditions. These results have been added in Supplementary tables.

9) In the optimization studies, how was reflux maintained without loss of solvent in an open test tube? An experimental for these optimization studies would be helpful.

Response: We use a very long test tube (25 cm) and there is almost no loss of solvent under our conditions. We have noted this in the revised SI file.

10) In a number of the HPLC traces of enantioenriched compounds, there is more than one minor peak near the retention time from the racemic trace (page 130, 166, 167, 188). How did the researchers know which peak to integrate? A co-injection of racemic and enantioenriched compound may be necessary to correctly assign the minor peak in these cases.

Response: We have re-prepared and purified these products and retested them on HPLC.

11) On page 183, the labels for the traces are confusing. It is recommended that the reaction 5-membered ring + catalyst = 7-membered ring is shown (not just reagents). Also, the HPLC trace of the racemic 7-membered ring should be added to this page for easy reference.

Response: Thanks for providing this suggestion. We have added full chemical equations and the HPLC trace of the racemic 7-membered ring product.

Referee 2:

1) Is this reaction applicable with other nucleophiles such as arylboronic acid?

Response: Under the current reaction conditions, our reaction is not compatible with arylboronic acid. New conditions may be developed in the future.

2) Why this nickel-catalyzed reaction shows high selectivity of 1,2-addition rather than 1,4-addition (of rhodium catalyst)? Is this selectivity controlled (or chanted) by the ligand? Or just nickel has tendency to undergo 1,2-addition?

Response: We have tried rhodium catalyzed asymmetric additions of alkenylboronic acid to alkenyl-substituted ketimines (**4a** and **2a**). The result showed that the reaction is a 1,4-addition. We consider nickel to be a "hard" Lewis acid and that it tends to coordinates with nitrogen. Conversely, rhodium is a "soft" Lewis acid and tends to coordinate with the C=C double bond, which leads to different results.

Of course, the ligand may change the property of the metal ion and thus change the regioselectivity. However, for now, we have found that only the BOX ligand can promote this Ni(II)-catalyzed addition reaction. New conditions may be developed in the future.

3) There are many typos. When submitting the manuscript, the authors should be more careful.

Response: The manuscript has been checked carefully before resubmission.

Reviewer #1 (Remarks to the Author):

In this revised manuscript, the authors have addressed my primary concerns regarding their proposed mechanism, as well as my more minor suggestions for revision.

For the mechanism discussion, they have much more thoughtfully described their hypothesis, and their discussion of the hindered bond rotation around C3-C4 is much more thorough. I am not an expert in computational studies and am unfamiliar with the WB97XD method, so I cannot comment on its suitability. However, the 6-31G(d,p) basis set is a commonly used level of theory for these types of calculations. I recommend that this manuscript be accepted after the computational discussion in the SI has been revised as follows:

Bond rotation around C1-C2 or C2-C3 would not be expected, because these bonds are part of the conjugated allyl cation fragment (not single bonds). The discussion around these possibilities obfuscates the key discussion around the possible bond rotation around C3-C4. I recommend that Supplementary Figure 8 be separated into two schemes. The first should be a reaction coordinate diagram comparing direct cyclization of the allyl cation SI-1 to (S)-5a vs. C3-C4 bond rotation and cyclization to (R)-5a. The relative energy levels of intermediates and ground states, as well as the calculated structures should be included. (Note: no calculated structures are shown, making it difficult to assess the validity of the computational studies.) The discussion of C1-C2 and C2-C3 bond rotation should be discussed separately, as these are far less likely. In addition, data for the calculated structures (intermediates and transition states) should be included in the SI. This should include atomic coordinates, ZPE's, and frequencies (particularly important for transition state structures). The inclusion of this data, as well as the discussion, greatly bolsters the authors' mechanistic hypothesis and strengthens the paper substantially.

A very minor note: "carbon cation" should be replaced with "carbocation" throughout the manuscript and SI.

Reviewer #3 (Remarks to the Author):

Computational analysis of some mechanistic aspects of the described reaction are performed using standard method and basis set, including solvent effects at SMD level. Computational level employed seems to be adequate to analyse this process. Authors must specify whether the energy values given are potential of Gibbs energies. According to supplementary Table 3, I guess they are Gibbs energies, but it is not clear.

The authors investigate several reaction pathways starting from SI-1 intermediate analysing the rotation around several bonds, C1-C2, C2-C3 and C3-C4. Their results support the fact that the ring expansion step is stereospecific. My main worry is to know whether the intermediate they are considering, SI-1 is the proper one to make such analysis. It does not correspond to the one shown in Figure 8 of the main text (the orientation of the styrenil group is not the same). They should employ the same geometrical arrangement shown in Figure 8 of the main text for their analysis (or at least

evaluate the relative energy between such intermediate and the one employed, SI-1).

I must say that that in their scan around C2-C3 bond starting from intermediate SI-1, calculations indicate that it connect to reactant SI-6 (an equivalent of intermediate 7 in Fig. 8). This suggests that the analysis may be fine. I get the impression that if the analysis is performed on the carbocation intermediate shown in Figure 8, the conclusions will not be significantly different to the one already described. Nevertheless, the structure selected for the analysis, SI-1, may be not the most adequate one; (in addition, the scans shown in supplementary Figures 9 and 10 does not show such intermediate as a minimum in the curve).

The graphs in supplementary Fig. 9 and 10 should be properly done: y axis, preferably energy in kcal/mol; x axis, they should show the values of the dihedral angles, and the atoms selected to define such dihedral angle.

Response to Referees

Mar. 31, 2018

Dear Editor,

We wish to submit the revised manuscript (tracking number: NCOMMS-17-29760A) entitled “*Ni(II)-Catalyzed Asymmetric Alkenylations of Ketimines*” for the publication in *Nature Communications*.

In this communication, we have developed a general enantioselective addition of alkenylboronic acids to ketimines, which represents the first example of a Ni(II)-catalytic system for such reactions. A relatively wide substrate scope of cyclic ketimines and alkenylboronic acids are compatible with our reaction conditions. The desired chiral α -tertiary allylic amines can be obtained in good yields (up to 99%) and enantioselectivities (up to >99% ee). A Ni(II)-catalyzed asymmetric cascade alkenylation/ring-expansion reaction of alkenyl cyclic ketimines was also developed. A series of seven-membered chiral sulfamides were synthesized with good yields (up to 94%) and enantioselectivities (up to >99% ee) under mild reaction conditions. Mechanistic studies showed that the alkenylation is the enantioselectivity-determining step, while the ring expansion step is a stereospecific process. The site-selectivity of the ring rearrangement expansion can be controlled by the formation of a large π -conjugated system or by steric interactions. Transformations of products for both types of reactions were conducted to show the potential applications of our methods.

I hope that the significance of our work satisfies the criteria as a communication for publication in *Nature Communications* and I am looking forward to hearing positive response from you regarding our manuscript.

Thank you very much for your work on our manuscript. The modifications for our manuscript are as follows:

Referee 1:

- 1) Bond rotation around C1-C2 or C2-C3 would not be expected, because these bonds are part of the conjugated allyl cation fragment (not single bonds). The discussion around these possibilities obfuscates the key discussion around the possible bond rotation around C3-C4. I recommend that Supplementary Figure 8 be separated into two schemes. The first should be a reaction coordinate diagram comparing direct cyclization of the allyl cation SI-1 to (S)-5a vs. C3-C4 bond rotation and cyclization to (R)-5a. The relative energy levels of intermediates and ground states, as well as the calculated structures should be included. (Note: no calculated structures are shown, making it difficult to assess the validity of the computational studies.) The discussion of C1-C2 and C2-C3 bond rotation should be discussed separately, as these are far less likely. In addition, data for the calculated structures (intermediates and transition states) should be included in the SI. This should include atomic coordinates, ZPEs, and frequencies (particularly important for transition state structures). The inclusion of this data, as well as the discussion, greatly bolsters the authors mechanistic hypothesis and strengthens the paper

substantially.

Response: Thanks for the reviewer's advice. We have separated Supplementary Figure 8 into two schemes (Supplementary Figure 8 and 9 in the revised manuscript). Data for intermediates and transition states have been added. Atomic coordinates and frequencies for transition state structures were shown at the end of SI (before Supplementary References).

2) A very minor note: carbon cation should be replaced with carbocation throughout the manuscript and SI.

Response: Thanks for the reviewer. We have changed "carbon cation" to "carbocation" throughout the manuscript and SI.

Referee 3:

1) Authors must specify whether the energy values given are potential of Gibbs energies. According to supplementary Table 3, I guess they are Gibbs energies, but it is not clear.

Response: The energy values given are potential of Gibbs energies. We have added notes at Supplementary Figure 8 and 9 in revised manuscript.

2) My main worry is to know whether the intermediate they are considering, SI-1 is the proper one to make such analysis. It does not correspond to the one shown in Figure 8 of the main text (the orientation of the styrenyl group is not the same). They should employ the same geometrical arrangement shown in Figure 8 of the main text for their analysis (or at least evaluate the relative energy between such intermediate and the one employed, SI-1).

Response: Thanks very much for giving this suggestion. We are sorry for making this reviewer to be confusing. It is our fault that we had forgotten to revise the structures in Figure 8 in the main text of the previous manuscript. The calculation results for the C-N bond breakage process showed that only this configuration could be formed, as shown in intermediate **SI-1** (SI-2 in revised manuscript). The original configuration in Figure 8 bears steric hindrance between catalyst and the styrenyl group, thus it is unfavored. This time, we have revised the configuration in Figure 8 of main text.

3) I must say that in their scan around C2-C3 bond starting from intermediate SI-1, calculations indicate that it connects to reactant SI-6 (an equivalent of intermediate 7 in Fig. 8). This suggests that the analysis may be fine. I get the impression that if the analysis is performed on the carbocation intermediate shown in Figure 8, the conclusions will not be significantly different to the one already described. Nevertheless, the structure selected for the analysis, SI-1, may be not the most adequate one; (in addition, the scans shown in supplementary Figures 9 and 10 does not show such intermediate as a minimum in the curve).

Response: As response for the second suggestion. For the question about "the scans shown in

supplementary Figures 9 and 10 does not show such intermediate as a minimum in the curve". This is because the starting intermediate (the first point) has not been structure-optimized. We have re-scan the rotation process using the structure-optimized intermediate SI-1 (SI-2 in revised manuscript). The Figures have been corrected and clearly showed that such intermediate is a minimum in the curve.

- 4) The graphs in supplementary Fig. 9 and 10 should be properly done: y axis, preferably energy in kcal/mol; x axis, they should show the values of the dihedral angles, and the atoms selected to define such dihedral angle.

Response: Thanks for giving this good suggestion. We have revised the graphs in supplementary Fig. 9 and 10 (Supplementary Figure 10 and 11 in revised manuscript.). The atoms selected to define the dihedral angles have been shown in corresponding structures.

Additionally, these changes have been marked in yellow color in the revised manuscript.

Thanks again for your and the referees' excellent work on our manuscript.

Yours sincerely,

Prof. Dr. Wanbin Zhang
School of Chemistry and Chemical Engineering
Shanghai Jiao Tong University
800 Dongchuan Road, Shanghai 200240, P. R. China
Tel: +86-21-5474-3265
Fax: +86-21-54743265
E-mail: wanbin@sjtu.edu.cn

Reviewer #3 (Remarks to the Author):

In the revised version of the manuscript the authors have addressed all the recommendations. My major concern about the selected structure as a key intermediate has now been clarified. They showed two different structures by mistake, but now are corrected and only intermediate SI-2 is shown. They now show the overall Gibbs energy barrier for the formation of SI-2 (31.2 kcal/mol) that is quite high (but the reaction is performed at 50 oC).

The graphs with the scan have been also corrected including values and units.

Regarding their analysis of rotation about bonds C1-C2, C2-C3 and C3-C4, they split the results in two different figures as suggested by reviewer 1. The presentation is now much more understandable. It is true that rotation about C1-C2 and C2-C3 is much more difficult due to conjugation. Authors made a mistake on Figure 9 legend because it should state C2-C3 instead of C3-C4.

Overall, I think the manuscript could be accepted.

Response to Referees

Apr. 24, 2018

Dear Editor,

We wish to submit the revised manuscript (tracking number: NCOMMS-17-29760A) entitled “*Ni(II)-Catalyzed Asymmetric Alkenylations of Ketimines*” for the publication in *Nature Communications*.

Thank you very much for your work on our manuscript. The following is our response to referee 3’s comments:

Referee 3:

- 1) In the revised version of the manuscript the authors have addressed all the recommendations. My major concern about the selected structure as a key intermediate has now been clarified. They showed two different structures by mistake, but now are corrected and only intermediate SI-2 is shown. They now show the overall Gibbs energy barrier for the formation of SI-2 (31.2 kcal/mol) that is quite high (but the reaction is performed at 50 °C). The graphs with the scan have been also corrected including values and units. Regarding their analysis of rotation about bonds C1-C2, C2-C3 and C3-C4, they split the results in two different figures as suggested by reviewer 1. The presentation is now much more understandable. It is true that rotation about C1-C2 and C2-C3 is much more difficult due to conjugation. Authors made a mistake on Figure 9 legend because it should state C2-C3 instead of C3-C4.

Response: Thanks very much for supplying these comments. We have changed the state in Supplementary Figure 9 from C3-C4 to C2-C3.

Thanks again for your and the referees’ excellent work on our manuscript.

Yours sincerely,

Prof. Dr. Wanbin Zhang
School of Chemistry and Chemical Engineering
Shanghai Jiao Tong University
800 Dongchuan Road, Shanghai 200240, P. R. China
Tel: +86-21-5474-3265
Fax: +86-21-54743265
E-mail: wanbin@sjtu.edu.cn